# Position: AI Must Become Planet-Centered, Not Just Human-Centered

**María Pérez-Ortiz** [1]

## Abstract

This position paper argues that contemporary AI paradigms are insufficient for supporting complex global goals and introduces Planet-Centered AI (PCAI) as a design philosophy and research agenda that reorients AI toward planetary-scale socio-ecological systems and their long-term trajectories. A planet-centered approach is grounded in systems thinking, treating Earth as an interconnected whole of which humans are part. We diagnose recurring limitations across AI frameworks—many of which remain human-centered—and show why these become especially consequential under current planetary conditions characterized by systemic risk, non-stationarity, and deep uncertainty. We then articulate how PCAI reshapes the AI lifecycle, from problem formulation and model design to evaluation and deployment, by emphasizing alignment with global agendas, developing system-aware AI foundations, trajectory-oriented evaluation, and monitorability. Finally, we advance a falsifiable claim: AI systems optimized without explicit consideration of systemic consequences are more likely to exacerbate systemic instability than to mitigate it.

## 1. Introduction

Over the past decade, the AI community has developed a range of paradigms to address the ethical, social, and technical risks of AI. Frameworks such as Human-Centered AI, Responsible AI, AI for Social Good/Sustainability, and AI safety have been essential in establishing that AI has far-reaching consequences, and that potential harms and broader impacts should inform algorithmic design and deployment.

Despite this progress, **this position paper argues that these paradigms are insufficient for enabling AI to mean-** ingfully support societies in confronting complex challenges: ML must be reoriented toward a planet-centered paradigm that treats systemic risk, long-term impact, and global goals as first-order design objectives.

As the world enters what is described as a polycrisis ([Lawrence et al.](), 2024), risks arise not from isolated failures but from coupled systems whose interactions generate self-reinforcing and systemic dynamics. Consider the following example: rising ocean temperatures alter predator-prey dynamics, enabling jellyfish blooms that clog coastal power plant cooling intakes, triggering forced shutdowns and energy instability ([Purcell et al.](), 2007). This is systemic risk, risks emerge from the coupling between systems. Similarly, climate change has been shown to aggravate more than half of known infectious diseases, through over a thousand distinct pathways linking physical, ecological, and social systems ([Mora et al.](), 2022). In such settings, feedback loops, nonlinear interactions, and path dependence—where early interventions shape and constrain future outcomes—can amplify risks and lock societies into trajectories that are difficult to reverse ([Delannoy et al.](), 2025; [Steffen et al.](), 2018). Recent work shows that AI is increasingly entangled with polycrisis dynamics through material pathways (e.g., energy use, resource extraction, infrastructure lock-in) and informational pathways (e.g., shaping behavior, accelerating decision cycles, synchronizing systems), with the potential to intensify systemic instability ([Creutzig et al.](), 2022).

It is precisely this systems perspective ([Meadows](), 2008)—focused on feedbacks, interactions, and trajectories—that remains absent from AI frameworks ([Kondor et al.](), 2024). We argue that AI methods are poorly suited to supporting planetary challenges, which exhibit properties of "wicked problems" ([Rittel & Webber](), 1973): e.g. non-stationarity and feedback-driven dynamics. This mismatch matters because such conditions increasingly characterize high-stakes domains in which AI is deployed, such as climate governance, technology regulation, and public policy ([Ilcic et al.](), 2025). Due to this misalignment, AI can generate systemic risks by interacting with or amplifying underlying system dynamics in unintended ways ([Schön et al.](), 2025). Yet frameworks for anticipating and evaluating systemic effects remain limited, leaving concepts of systemic risk underdeveloped and inconsistently operationalized in AI governance ([Carey](), 2025; [Stahl et al.](), 2023).

[1] Centre for AI, Department of Computer Science, University College London, London, UK. Correspondence to: María Pérez-Ortiz <maria.perez@ucl.ac.uk>.

*Proceedings of the 43rd International Conference on Machine Learning*, Seoul, South Korea. PMLR 306, 2026. Copyright 2026 by the author(s).

Climate change illustrates why this systems perspective matters: The scale of global warming is driven not only by human emissions, but by reinforcing feedbacks within the Earth system. Feedbacks, such as water-vapor amplification and cloud responses, roughly double to triple the temperature response to anthropogenic greenhouse gas emissions, accounting for much of the 1.2°C of global warming observed to date (IPCC, 2013). Similar dynamics arise in technological systems (Galaz et al., 2021; Ilcic et al., 2025), where early design choices, deployment incentives, and governance can entrench behaviors that persist even as cumulative harms become evident. In both cases, effective intervention depends on understanding feedbacks, long-term dynamics, and systemic interactions (Stirling, 2010)—dimensions that current AI paradigms struggle to represent.

We propose Planet-Centered AI (PCAI) as a design philosophy and research agenda that complements the limits of human-centered framings. Human-centered approaches rightly focus on protecting individuals from harm, but often do not consider environmental risks, long-term dynamics, and systemic effects. Planet-centered approaches instead recognize these as constitutive of human and planetary futures. PCAI expands responsibility beyond users, communities and societies to include ecosystems, systemic risks, and Earth-system dynamics, reframing intelligence as a tool for collective understanding and planetary stewardship.

## 2. The Limits of Contemporary AI Paradigms in the Anthropocene

Across the AI ethics landscape, a strong commitment to protecting humanity has emerged (Jobin et al., 2019). Frameworks such as Human-Centered AI (HCAI) (Shneiderman, 2020) and Responsible AI have expanded research beyond narrow definitions of performance (Schmager et al., 2025), introducing desiderata such as explainability, human oversight, robustness and fairness, as well as a move towards human augmentation. We argue, however, that these paradigms remain insufficient in the Anthropocene (Creutzig et al., 2022). The Anthropocene denotes the geological era in which human activity—increasingly mediated by technology—has become the dominant force shaping Earth systems. It is characterized by global interconnectedness, nonlinear change, tightly coupled dynamics, and amplified systemic risk—dynamics increasingly accelerated by AI (Delannoy et al., 2025; Galaz et al., 2021).

### 2.1. Wicked problems in the Anthropocene

A central source of AI's limitations lies in the distinction between *tame* and *wicked* problems, originally introduced to explain why scientific and engineering approaches often fail in complex social and policy domains (Rittel & Webber, 1973). Tame problems—such as puzzles or well-defined optimization tasks—have stable objectives, agreed-upon problem formulations, and objective criteria for success. Even when technically complex, they can be decomposed, optimized, and evaluated against fixed goals.

Wicked problems exhibit properties that violate these assumptions[1] (Peters, 2017): Objectives are contested and non-stationary; interventions alter system dynamics; effects propagate across domains; and outcomes unfold over long, uncertain horizons. There is no well-defined global optimum, no safe regime for trial-and-error learning, and no reliable evaluation of success. Examples include climate change mitigation and adaptation, biodiversity conservation, sustainability transitions, and poverty reduction, where interventions interact with social, economic, and ecological systems (Toyama, 2010). Problems are further characterized by deep uncertainty, where key elements of the system—causal structure, feedbacks, objectives, or future conditions—are unknown, contested, or not reliably quantifiable (Marchau et al., 2019). Anthropocene challenges specifically, are frequently described as *super-wicked* because they intensify these features: decisions are high-stakes and potentially irreversible, feedbacks amplify over time, and action must occur under deep uncertainty and time pressure.

### 2.2. AI Failure Mechanisms in the Anthropocene

**We argue that the properties of Anthropocene's challenges stand in tension with the assumptions in AI, and this mismatch drives the unintended and systemic consequences of AI.** Next, we examine existing practice, highlighting mechanisms for AI failure in wicked contexts.

#### 2.2.1. TECHNICAL MISALIGNMENT

Technical misalignment refers to the failure that arises when AI — e.g. built around assumptions of stationarity, sample independence, and closed-world formulations — is deployed in wicked contexts whose dynamics structurally violate these conditions (Figure 1; see Appendix A and Table 2 for a diagnostic). These assumptions enable optimization, benchmarking, and iterative improvement in tame domains, and not all sustainability problems are wicked: coordinating the orientation of wind turbines in a wind farm to maximise energy output, for instance, involves a clear objective, stable system boundaries, and directly measurable success (Howland et al., 2022). When similar methods are applied to wicked challenges, however, this misalignment becomes consequential. AI-based biodiversity monitoring for conservation operates within entangled social–ecological systems where objectives are contested and boundaries are porous. Such interventions reshape behaviour — altering land use, enforcement practices, conflict dynamics, and surveillance

---

[1]Appendix A provides a diagnostic for wickedness and maps wicked system properties to the AI assumptions they violate.

| Wicked problems | Properties of wicked systems | Mechanisms for AI failure in wicked systems |
|---|---|---|
| 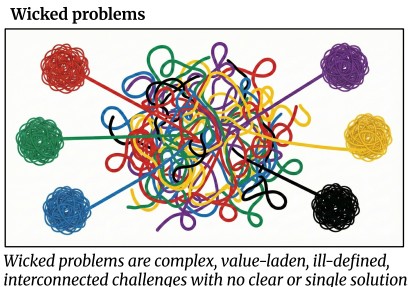 | Ambiguity   Non-stationarity

Emergence   Fragility

Trade-offs   Contestedness

Path-dependence   Coupling

Reflexivity   Uncertainty | The mechanisms below drive AI's unintended and systemic consequences in wicked systems.

• **AI is technically misaligned** → AI assumptions fail.
• **Narrow framing of value and progress** → The wrong objectives are optimised.
• **Impact blind spots** → Consequences go unnoticed.
• **Risk amplification** → Harms scale.
• **Absence of shared direction** → Failures persist. |
| *Wicked problems are complex, value-laden, ill-defined, interconnected challenges with no clear or single solution.*

*They show up everywhere — in social policy, technology, environment, and large-scale change efforts.* | | |

*Figure 1.* Wicked problems, their structural sources of difficulty, and the mechanisms of AI failure.

relationships — while introducing privacy risks (Duffy et al., 2019; Sandbrook et al., 2021). Even well-intentioned deployments can mis-steer system trajectories because the learning problem becomes ill-posed and emergent consequences — arising from component interactions rather than any single component — fall outside task-level evaluation.

### 2.2.2. NARROW FRAMING OF VALUE AND PROGRESS

Notions of progress are implicitly encoded in AI: performance is typically measured through gains in efficiency, accuracy, scale, or generality (Pansera & Fressoli, 2021; Birhane et al., 2022), embedding a conception of advancement that equates progress with optimisation and expansion. Empirical analyses show that influential AI research overwhelmingly prioritises quantitative performance gains, while explicit articulation of societal benefits or harms remains rare (Birhane et al., 2022). This framing aligns with market-driven theories of development in which productivity and growth are proxies for social value (Pansera & Fressoli, 2021) — assumptions that are problematic in coupled socio-ecological systems, where AI may register as successful while contributing to inequality, fragility, or environmental degradation (Schön et al., 2025). Alternative perspectives emphasise distributional fairness, resilience, human capabilities, and compatibility with ecological limits (Pansera & Fressoli, 2021; Kallis et al., 2025) — criteria largely absent across the AI lifecycle.

At the same time, many AI governance frameworks articulate high-level value aspirations—such as human-centeredness or social good—without specifying how these should be operationalized (Jobin et al., 2019; Whittlestone et al., 2019). This underspecification ultimately defers difficult questions about trade-offs and responsibility (Mittelstadt, 2019). Core concepts such as "the human" or "the good" remain ambiguous: humans may be implicitly treated as users, consumers or workers, despite the fact that these roles entail incompatible values and consequences (Bucknall & Dori-Hacohen, 2022; Selbst et al., 2019). As a result,

moral aspirations are translated into technical objectives through problem formulations, proxies, and metrics that embed implicit value trade-offs without sustained scrutiny (Birhane et al., 2022; LaCroix & Luccioni, 2025). Social media recommendation illustrates this. These systems often define progress through engagement metrics —clicks, shares, dwell time—and succeed by those measures. However, engagement as a proxy equates what users interact with to what they want. An algorithmic audit of Twitter/X found that the platform's engagement-based ranking algorithm amplifies emotionally charged and out-group hostile content, and importantly, that users do not prefer the political content selected by the algorithm (Milli et al., 2025).

### 2.2.3. IMPACT BLIND SPOTS IN ENTANGLED SYSTEMS

A consistent finding across AI governance and ethics is that impact assessment is narrowly scoped and weakly integrated across the lifecycle (Stahl et al., 2023; UNESCO, 2023): systematic reviews highlight the absence of methods for anticipating indirect, cumulative, long-horizon, and intergenerational effects (Stahl et al., 2023; Kondor et al., 2024), with evaluation typically confined to tasks, models, or deployment settings (Ahlborg et al., 2019). In coupled socio-ecological systems, this produces two types of blind spots (Schön et al., 2025): risks from entanglement, where consequences propagate across system boundaries through couplings invisible to single-domain evaluation; and risks from intervention depth, where optimising within existing dynamics leaves the feedback structures driving undesirable trajectories intact (Meadows, 2008). AI-driven precision agriculture illustrates both: Tools such as pesticide-reduction drones evaluated on chemical-load reduction succeed within field-level environmental metrics, yet the technology is largely inaccessible to smallholders, accelerating consolidation toward the monoculture model that high pesticide use is structurally bound up with (Altieri et al., 2024) — entangling food security, rural livelihoods, and agrobiodiversity in consequences no environmental metric was designed to detect, while leaving drivers of agricultural harm intact.

Anthropocentric framing constitutes an important instance of such blind spots. Empirical analyses show that only 16–26% of AI ethics guidelines explicitly address non-human life, environmental sustainability, or ecological systems (Sebestyén, 2025; Rigley et al., 2023; Jobin et al., 2019). Where these concerns appear, non-human entities and planetary processes are typically treated as externalities, valued primarily for their instrumental role in human well-being. However, harms such as biodiversity loss and the destabilization of life-support systems unfold gradually, interact with other stressors, and manifest over long timescales (Rigley et al., 2023; Bucknall & Dori-Hacohen, 2022). While such impacts may not register as direct or near-term risks now, they accumulate and compound, constraining the conditions for both human and non-human communities to persist and flourish (Bucknall & Dori-Hacohen, 2022).

### 2.2.4. RISK AMPLIFICATION

Technical misalignment, obscured by impact blind spots and reinforced by narrow framings, does not merely limit AI's effectiveness—it can amplify systemic risk (Schön et al., 2025): A clear example of this mechanism is provided by rebound effects in sustainability. AI for Sustainability has shown that AI can improve efficiency in tasks related to energy, agriculture, and transport (Gohr et al., 2025). However, these gains are typically evaluated within narrow system boundaries that exclude behavioral, economic, and institutional responses. As a result, efficiency improvements can lower costs, accelerate adoption, and expand overall system activity, offsetting—or even reversing—environmental benefits (Wright et al., 2025; Mhlanga, 2025). Autonomous vehicles are a good example. Autonomous driving is designed to improve safety and reduce per-mile emissions through optimised routing and driving efficiency, objectives that are human-centered and environmentally motivated. At the vehicle level, autonomy introduces an average 21% decrease in operational emissions through improved fuel economy (Onat et al., 2023). However, by reducing the perceived cost and inconvenience of travel, autonomous vehicles stimulate additional demand—through longer commutes, modal shifts away from public transit, and empty vehicle repositioning miles. Estimates of induced travel demand range from 2% to 47% increases in household vehicle miles travelled (Taiebat et al., 2019). When the full lifecycle is considered (including manufacturing, increased vehicle use, and infrastructure expansion) autonomous electric vehicles may emit approximately 8% more greenhouse gas emissions than their non-autonomous counterparts (Onat et al., 2023). Similar dynamics appear in Sustainable AI: model efficiency gains do not account for downstream effects such as wider deployment, increased demand, or infrastructure expansion, which at scale lock systems into trajectories that could intensify resource use and constrain future options (Wright et al.,

2025). From a strong sustainability perspective (Neumayer, 2010), this is particularly concerning because certain ecological functions are subject to absolute limits and cannot be substituted through efficiency alone.

Beyond rebound effects, AI deployment introduces additional amplification pathways: i) its pervasiveness across critical infrastructures; ii) a pace and scale that outstrip regulatory capacity; iii) its technical opacity, which limits democratic oversight; and iv) propagation risks, where reliance on the same models and datasets allows localized failures to cascade (Galaz et al., 2021).

### 2.2.5. ABSENCE OF SHARED GLOBAL DIRECTION

A deeper limitation lies in the absence of a widely shared agenda guiding AI research, deployment, and evaluation (Carey, 2025; Whittlestone et al., 2019; Hagendorff, 2020). In practice, problem selection is shaped by data availability, benchmarkability, and short-term incentives, favoring domains that integrate smoothly into existing pipelines (Gohr et al., 2025). In environmental applications, this is reflected in the dominance of satellite imagery, while less observable but ecologically critical processes (e.g. soil health or biodiversity interactions) remain underexplored (Gohr et al., 2025). Broadly, a systematic review of 792 articles at the intersection of AI and the Sustainable Development Goals (SDGs) finds that very few effectively bridge advanced AI with deep sustainability expertise, and that the literature remains fragmented into silos dominated by forecasting and system optimisation (Gohr et al., 2025).

As a result, AI systems tend to model what is observable and measurable, rather than identifying leverage points, i.e. the small set of structural interventions that drive system-wide impact. A sensitivity analysis of the Earth4All world model illustrates the importance of identifying high-leverage interventions in coupled systems: of 21 global policies originally proposed to deliver planetary wellbeing (Stoknes et al., 2025), just six account for most of the system-level improvement, with three turnarounds — energy, inequality, and poverty — dominating outcomes across all indicators (Crescenzi et al., 2024). Unlike domains guided by shared global aspirations—such as the SDGs—AI development remains fragmented across sectors and jurisdictions. Existing international efforts (e.g. Global Digital Compact and emerging UN-level AI governance) acknowledge the need for alignment but offer limited guidance on research priorities and acceptable trade-offs (UNESCO, 2023). In the absence of direction, AI development defaults to optimizable objectives, reinforcing the persistence of misalignment, risk amplification, and narrow value framing. This represents a missed opportunity to orient AI research toward the complex, high-stakes challenges of the Anthropocene (Creutzig et al., 2022).

# 3. Planet-Centered AI

Our previous analysis showed that AI paradigms struggle in Anthropocene contexts. These limitations indicate that existing notions of responsible AI are insufficient under current planetary conditions. Planet-Centered AI (PCAI) is proposed as a response: **PCAI is a design philosophy and research agenda that integrates systems thinking and ecological responsibility across the AI lifecycle, aligning AI development with the demands of planetary-scale challenges.** By ecological responsibility, PCAI means treating the integrity, resilience, and limits of ecological systems as first-class design constraints—rather than as externalities inferred indirectly through human outcomes. Rather than asking how AI can become more capable, efficient or ethical in isolation, PCAI poses a different orienting question: *How can AI support societies in understanding, navigating, and responsibly shaping Earth-system futures?* Figure 3 summarizes PCAI design principles. The remainder of this section shows how these principles reshape the AI lifecycle across: (i) foundational research priorities, and (ii) applied design requirements for real-world AI systems in planetary contexts. Importantly, PCAI does not imply uniform requirements across all research; like HCAI, the depth of engagement should scale with the degree of wickedness and the potential for irreversible or systemic harm (Shneiderman, 2020). The four quadrants of Figure 2 operationalise this scaling: tame problems follow standard AI practice; value-contested problems require system mapping and Pareto-style evaluation; knowledge-deficient problems require foresight and robustness under deep uncertainty; and fully wicked problems require the full PCAI lifecycle including monitorability and revisability. Table 1 translates these principles into concrete actions across researchers, institutions, funders, and governance contexts.

## 3.1. Problem Setting & Diagnosis

Given the urgency and complexity of planetary challenges, **under PCAI both foundational and applied ML research should be oriented toward shared goals**, such as those articulated in national or international agendas. This requires an expert-led assessment of where AI can provide the greatest leverage relative to alternative interventions, what solutions may multisolve challenges[2], and which technical constraints prevent AI from contributing meaningfully. These constraints may be methodological (e.g., uncertainty quantification, causal reasoning, human–AI interaction), transdisciplinary (e.g., system-level evaluation frameworks that capture cross-domain effects), or pre-conditional for

---

[2]For example, Project Drawdown (Hawken, 2017)—one of the most comprehensive, evidence-based assessments of climate mitigation strategies—ranks educating girls among the most effective interventions for reducing global greenhouse gas emissions, highlighting the impact of social factors in environmental challenges.

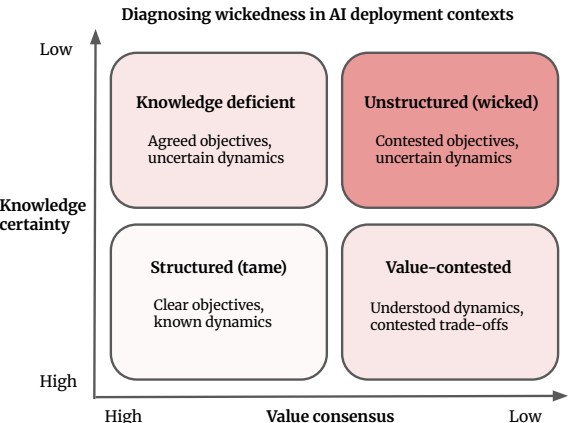

**Diagnosing wickedness in AI deployment contexts**

*Figure 2.* Wicked problem classification adapted from (Hisschemöller & Hoppe, 2018). Problems vary in the degree of wickedness along two dimensions: whether the system's causal dynamics are understood (knowledge certainty) and whether there is agreement on objectives and trade-offs (value consensus). As wickedness increases, more stages of the PCAI lifecycle become relevant. Wickedness is further amplified by coupling to adjacent socio-ecological systems, which can introduce additional uncertainties and contestations in any quadrant. Appendix A includes further information on the diagnostic and worked examples.

applied science (e.g., missing data, inadequate monitoring infrastructure, weak interfaces through which model outputs influence decisions). Such assessments will guide application choice, but also, importantly, shape technological development agendas, directing research toward the foundations for AI to contribute meaningfully to planetary goals.

For planetary goals, **PCAI introduces system mapping and theories of change as design preconditions for applied AI research and real-world deployment**[3]. PCAI requires AI interventions to be situated within the complex systems in which they operate. System mapping makes explicit the relevant system boundaries and dynamics, decision-makers, affected communities, and temporal horizons over which impacts unfold—supporting a diagnosis of which wicked characteristics the problem exhibits. This mapping reduces the risk that consequential dynamics are omitted during problem formulation and evaluation. PCAI further requires that criteria for success be articulated *before* modeling begins, grounding optimization targets and evaluation metrics in system-level effects. Researchers are expected to document a causal account of how model outputs should influence decisions and system dynamics—that is, a theory of change. This will identify intended leverage points, plausible feedbacks, and potential failure modes—such as behavioral adaptation or rebound effects—that could undermine intended benefits. This approach mirrors anticipatory reasoning in public policy design and provides a basis for

---

[3]We illustrate this with a conservation use case in Appendix C.

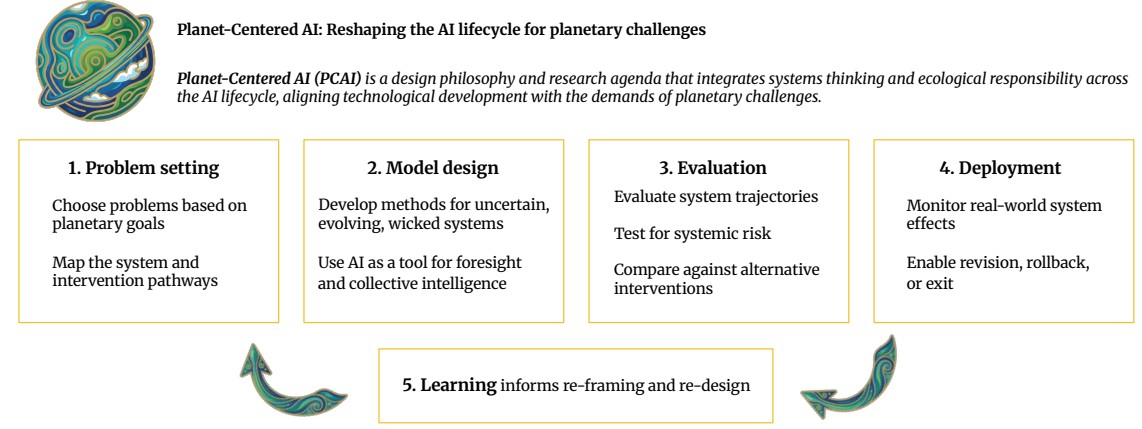

**Planet-Centered AI: Reshaping the AI lifecycle for planetary challenges**

*Planet-Centered AI (PCAI) is a design philosophy and research agenda that integrates systems thinking and ecological responsibility across the AI lifecycle, aligning technological development with the demands of planetary challenges.*

| 1. Problem setting | 2. Model design | 3. Evaluation | 4. Deployment |
|---|---|---|---|
| Choose problems based on planetary goals | Develop methods for uncertain, evolving, wicked systems | Evaluate system trajectories | Monitor real-world system effects |
| Map the system and intervention pathways | Use AI as a tool for foresight and collective intelligence | Test for systemic risk | Enable revision, rollback, or exit |
| | | Compare against alternative interventions | |

**5. Learning** informs re-framing and re-design

*Figure 3.* PCAI key design commitments across the lifecycle. Each stage emphasizes anticipating system effects and trajectories.

evaluating the wickedness of the problem (Peters, 2017).

## 3.2. Model Design

**PCAI strengthens systems-aware technical foundations.** PCAI aims to reorient AI research in light of the failure mechanisms identified. Many relevant challenges are already recognized within the AI community, but are typically addressed in isolation or without explicit attention to system dynamics. Consider non-stationarity. While work on continual learning and distribution shift focuses on maintaining model performance as data changes, planetary systems often exhibit endogenous change, abrupt regime shifts, and tipping points that invalidate assumptions of gradual or reversible dynamics. Under PCAI, non-stationarity therefore raises questions not only of adaptation, but of evaluation: models must be stress-tested against plausible alternative system regimes and structural breaks (Beucler et al., 2024), rather than optimized for a single expected distribution. This shift motivates greater emphasis on distributionally robust and minimax-regret formulations that aim to avoid catastrophic failure under deep uncertainty, rather than maximizing average-case performance — though these formulations may carry real costs in task-level efficiency. The Pareto framework in Section 3.3 is designed to surface these trade-offs. Uncertainty provides a second example. AI focuses on aleatoric and epistemic uncertainty, yet planetary contexts are frequently characterized by deep uncertainty. Under PCAI, identifying, communicating, and reasoning under such uncertainty becomes a core technical requirement. Related research on open-endedness begins to address novelty and unanticipated conditions (Stanley, 2019), but remains underdeveloped. At the foundational level, PCAI motivates a research agenda that prioritizes robustness, adaptability, and uncertainty-aware reasoning. At the applied level, researchers are expected to draw on prior diagnosis of system wickedness to inform model selection, training objectives, evaluation protocols, and deployment strategies.

**PCAI emphasizes AI as epistemic infrastructure.** Given the properties of wicked systems, automated decision-making is brittle, and prediction — while useful for short-horizon forecasting — can exacerbate the failures identified in Section 2 by collapsing deep uncertainty into a single expected projection. In complex systems, such overconfident forecasts obscure alternative futures and intervention pathways (Amoore, 2023; Pérez-Ortiz, 2024; Søgaard Jørgensen et al., 2024). Instead, PCAI builds on a longstanding tradition of using computation to augment human reasoning in complex systems (Meadows et al., 1972; Selin et al., 2023; Van Beek et al., 2020) — through simulation, exploratory modeling, and scenario analysis (Lavin et al., 2021; Pérez-Ortiz et al., 2022) — reframing AI's role from prediction and control toward foresight (Pérez-Ortiz, 2024; Bankes, 1993). The goal of foresight is epistemic: to generate understanding about how complex systems behave, evolve, and respond to intervention (Selin et al., 2023), supporting sensemaking and collective reasoning rather than automated decision-making. Human–AI interfaces are therefore oriented toward deliberation, contestation, and coordinated judgment.[4] Integrated Assessment Models (IAMs) illustrate both the need for and the feasibility of this reorientation. IAMs have long supported climate governance by linking physical, economic, and social dynamics to explore transformation pathways (Van Beek et al., 2020), yet most still rely on opaque optimisation solvers that converge on a single expected trajectory, collapsing deep uncertainty and contested values into a fixed objective. Emerging work is shifting IAMs toward foresight-oriented architectures: interpretable multi-agent reinforcement learning enables exploration of cooperative strategies across heterogeneous actors with competing interests (Rudd-Jones et al., 2025; Biswas

---

[4]Relevant insights include trends, emerging risks, trade-offs, black swans, gray rhinos, tipping points, value mappings, causal loops, cross-impact relations, and other systemic features (Selin et al., 2023; Pérez-Ortiz, 2024).

et al., 2025), while mixture-of-experts frameworks couple IAMs with agent-based and Earth-system models to test policy robustness across scales and structural assumptions (Filatova et al., 2025). Together, these approaches instantiate foresight as a technical response to full wickedness: mapping the space of possible futures supports deliberation over contested values, while multi-objective formulations, robustness testing across structural assumptions, and world models address deep uncertainty — the two dimensions that define the hardest challenges. Climate change is the canonical case: Earth-system dynamics are so deeply entangled with human actions through energy, food, and economic systems that the challenge is not to predict a single trajectory for Earth's temperature but to understand how different interventions lead to different possible futures — so that societies can reason about what responsible action requires.

### 3.3. Evaluation

**PCAI reframes evaluation as a tool for anticipating system-level consequences rather than ranking models.** While standard metrics remain useful for task performance, they are insufficient for understanding how AI shapes system behavior once deployed in wicked contexts. PCAI therefore adopts an umbrella evaluation approach that complements task metrics with analyses of trade-offs, stability, and systemic risk. Rather than collapsing performance into a single score, PCAI emphasizes evaluation practices that make competing objectives explicit across possible system trajectories. Pareto frontiers are used to surface tensions between efficiency, equity, resilience, or environmental impact, supporting transparent deliberation over trade-offs. We refer to this approach as trajectory-oriented evaluation: these trade-offs may help assess how models may shape integrative system trajectories over time.

**PCAI introduces systemic risk probes**. Evaluation explicitly tests for amplification mechanisms e.g. rebound effects and correlated failures. Where possible, simulation-based analyses (Guliyeva et al., 2025) are used to explore how deployment may alter broader system dynamics.

**PCAI encourages counterfactual baselines**. Models are compared against state-of-the-art, but also against no-AI baselines, simple heuristics, or alternative non-ML interventions. This makes opportunity costs visible and avoids justifying deployment solely on benchmark gains.

Sound evaluation in wicked contexts is an open challenge — metrics will always be reductionist, proxies may be necessary, PCAI foregrounds this rather than resolving it.

### 3.4. Deployment

**PCAI reframes deployment around monitorability**. Deployment is understood as a sustained intervention in an evolving system. Monitoring therefore extends beyond model-level signals—such as prediction error or data drift—to encompass system-level responses, including behavioral adaptation, rebound dynamics, early warning signals of emerging patterns (e.g., black swans or gray rhinos), and distributional effects that may indicate the reinforcement of fragile trajectories.

**PCAI treats deployed systems as revisable**. Continued operation is provisional and contingent on observed system-level impacts. Deployment includes predefined escalation, modification, and rollback pathways, triggered by monitored risk indicators. This requires that system boundaries, assumptions, contexts of use, and intervention pathways are specified in advance, so that observed changes can be attributed, contested, and acted upon.

### 3.5. A Falsifiable Claim

Recent work on Anthropocene traps (Søgaard Jørgensen et al., 2024) characterizes many contemporary crises as self-reinforcing trajectories in which short-term gains or delayed feedbacks erode long-term system resilience (Steffen et al., 2018). These traps—such as growth dependence, infrastructure lock-in, and rebound dynamics—are not caused by any single technology, but are frequently intensified by technologies that accelerate scale, efficiency, or coordination without attention to system-level feedbacks. In this context, technology functions as a powerful modulator of system trajectories, capable of stabilizing or destabilizing social–ecological systems. Against this background, PCAI sets a contestable and empirically examinable hypothesis:

> *In domains governed by wicked dynamics, AI systems optimized for efficiency or narrow objectives—without explicit consideration of systemic feedbacks and long-horizon effects—are more likely to exacerbate systemic instability than to mitigate it.*

This is, AI systems optimized for speed, scale, or narrow task performance, can reshape system dynamics, accelerating feedback loops, triggering rebound effects, and reinforcing Anthropocene traps that mask accumulating risk while narrowing future options (Søgaard Jørgensen et al., 2024). Apparent short-term improvements may coincide with declining system resilience and shrinking safe operating space. This claim is falsifiable. It predicts that AI systems designed and evaluated under PCAI principles should exhibit measurably different system-level effects than conventional deployments. This claim should be empirically testable through stress-testing and systems monitoring: evidence that PCAI-aligned systems reduce rebound effects or expand the safe operating space of a system would support the paradigm; evidence that they do not would falsify it.

# 4. Alternative Views

## 4.1. Human-Centered AI (HCAI) as Sufficient

A common view holds that HCAI provides a sufficient framework for addressing planetary concerns, since environmental instability ultimately harms humans. From this perspective, the challenge is not anthropocentric framing but incomplete implementation, such as extending temporal horizons or improving impact assessment. PCAI agrees that human well-being depends on planetary stability, but argues that anthropocentric framings treat environmental and systemic risks only indirectly, as downstream proxies for human harm, which can delay risk recognition and weaken responses to emerging dynamics. Appendix B compares PCAI to HCAI along different relevant dimensions.

Influential assessments of AI's relationship to the SDGs illustrate this limitation. (Vinuesa et al., 2020) find that AI may act as an enabler on 79% of SDG targets but may inhibit 35%, yet the authors themselves acknowledge that the interactions between targets—where progress on one may undermine another—remain poorly characterised, and that "novel methodologies are required to ensure that the impact of new technologies are assessed from the points of view of efficiency, ethics, and sustainability". PCAI responds to this call by providing the system-level reasoning that target-by-target assessment cannot capture.

## 4.2. AI Safety as Dominant Risk Framework

A second view argues that AI safety and alignment research already addresses long-term and catastrophic risks, rendering additional planetary framing unnecessary. AI safety has indeed developed powerful tools for analyzing misalignment, loss of control, and other AI internal failure modes. PCAI agrees that this work is essential, but argues that it targets a different class of risks. AI safety focuses primarily on whether AI systems pursue intended objectives and remain controllable (Russell, 2019). By contrast, many relevant risks in the Anthropocene arise from the deployment of AI within socio-ecological systems, where even well-aligned systems can amplify existing crises. From a planetary perspective, the central concern is thus how AI reshapes system dynamics over time. PCAI therefore complements AI safety by shifting attention from internal alignment to system-level embedding: safety asks whether AI systems behave as intended, while PCAI asks whether those intentions, when enacted at scale, contribute to stable, resilient, and sustainable planetary trajectories (Steffen et al., 2018).

## 4.3. Systemic Reasoning Outside the Scope of AI

A related objection holds that planetary-scale dynamics, system mapping, and theories of change lie outside the scope of AI, belonging instead to policy or Earth system science.

From this view, AI should focus on general-purpose tools, leaving system-level reasoning to downstream users, with incremental improvements—such as better benchmarks, audits, or regulation—seen as sufficient. PCAI does not claim that AI researchers should model entire planetary systems or replace policy judgment. Rather, it argues that AI design choices inevitably encode assumptions about the system modeled itself. When left implicit, these assumptions can lead to failure modes which amplify risk. System diagnosis is therefore set as a design precondition for applied science rather than a modeling task: it constrains objective specification, evaluation, and deployment, often in collaboration with domain experts. In this sense, PCAI shifts the burden from individual researchers to interdisciplinary processes, motivating concrete technical commitments.

Field evidence supports this position. A recent review of 25 GenAI deployments across low and middle income countries found that a common thread was the shift from a tech-solutionist paradigm to a socio-technical approach, as "problems tended to be nested, with one issue revealing others, requiring adaptability and the application of a holistic perspective" (Adams et al., 2026). These practitioners—working in health, agriculture, education, and gender-based violence—did not set out to do system mapping, but discovered that deployment in wicked contexts demanded it. This suggests that systemic reasoning is not an optional add-on to AI design but an emergent requirement of deployment in complex settings.

## 4.4. Technological Progress as Primary Solution

A final view holds that technological innovation will mitigate planetary harm through decoupling, whereby efficiency gains allow growth without increasing environmental impact. PCAI acknowledges real efficiency gains enabled by AI, but argues that in coupled systems these gains often trigger rebound effects, scale expansion, and lock-in, amplifying systemic risk rather than reducing it. It is therefore crucial for AI paradigms to consider these effects.

Empirical evidence also challenges this alternative view. Decades of research on information technology for development have shown that technology acts as a multiplier on pre-existing conditions: it accelerates progress where the underlying dynamics are already favourable, but widens disparities where they are not (Toyama, 2011). The US poverty rate, for instance, has not declined since 1970 despite four decades of intensive digital innovation (Toyama, 2010). Scaling this observation to AI, PCAI argues that without deliberate reorientation of objectives and evaluation, more capable AI systems will amplify the trajectories societies are already on — including unsustainable ones.

*Table 1.* Call to Action: Operationalizing PCAI across the AI ecosystem

| Actor | Concrete Actions |
| --- | --- |
| **Foundational ML Researchers** | **Develop epistemic AI for foresight.** Advance methods that support collective sensemaking, e.g. exploratory modeling, scenario generation, stress-testing or world models. **Build AI paradigms for evolving complex systems.** Prioritize learning under non-stationarity, regime shifts, and deep uncertainty, including robustness to structural change. **Advance hybrid and collective intelligence.** Design human–AI interaction paradigms that support deliberation, co-creation, and coordinated judgment. |
| **Applied ML Researchers** | **Mandate explicit system mapping.** Situate AI interventions within complex systems. **Require explicit theories of change.** Articulate how outputs influence system dynamics. **Demonstrate comparative value to alternatives.** Specify opportunity costs and risks. |
| **Research Institutions & Funders** | **Institutionalize planetary agenda-setting.** Translate existing international and national goals into AI-relevant research priorities through expert-led, interdisciplinary panels. **Tie funding to agenda.** Require proposals to specify which targets within planetary goals they impact, why AI is an appropriate lever, and what technical constraints exist. **Support high-uncertainty, long-horizon planetary research.** Fund work with exploratory outcomes, long validation cycles, and non-benchmarkable success criteria. **Require comparative and counterfactual baselines.** Make funding contingent on comparisons to no-AI and alternative interventions to avoid default technological solutionism. |
| **Conferences, Journals, and the ML Community** | **Redefine scientific contribution.** Recognize system-level evaluation, robustness analysis, scenario stress-testing, and systemic risk probes as first-class research outputs. **Encourage transparency of assumptions and impacts.** Require reporting of system boundaries, theories of change, and anticipated downstream effects. **Create durable venues for planetary reasoning.** Establish tracks, review criteria, and workshops for AI-augmented foresight, long-term trajectories, and planetary stewardship. **Extend AI ethics standards to planetary responsibility.** Evolve ethical frameworks to include ecological impacts, planetary limits, and systemic risk. **Reduce reliance on single-score rankings.** Encourage Pareto-front, trade-off, and robustness-oriented evaluation. **Standardize system mapping artifacts,** e.g. developing an "Impact Datasheet". |
| **Education and Training** | **Treat systems reasoning as a core ML competency.** Integrate complex systems dynamics and systems impact assessment into AI curricula. **Teach system mapping for AI practice.** Train students and practitioners to map system boundaries, feedbacks, and causal pathways linking model outputs to systemic effects. |
| **Governance and Deployment Contexts** | **Extend existing impact assessment regimes.** Embed systemic risk analysis into lifecycle governance processes and develop tools for failure assessment. **Mandate system-level monitoring and independent verification.** Require observation of behavioral, ecological, and institutional responses to deployment. **Treat deployment as provisional and revisable.** Require predefined escalation, rollback, and decommissioning triggers tied to systemic risk indicators. |

## 5. Call to Action and the Path to PCAI

PCAI calls for action across the AI ecosystem—spanning research practice and incentives, evaluation norms, and governance—to better align AI development with planetary challenges. PCAI is intentionally ambitious. It does not offer a complete solution, nor does it claim that AI could fully model or control planetary systems. Instead, it delineates an initial scope for a long-term research agenda—one that may unfold over decades—aimed at equipping societies with AI and impact assessment tools that support foresight, deliberation, and responsible intervention in complex challenges. In this sense, PCAI invites the AI community to contribute its distinctive technical expertise to the defining challenges of the Anthropocene. Table 1 summarizes the core commitments of PCAI and outlines how they translate into concrete shifts across the AI lifecycle.

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

# A. Diagnosing wickedness and its tensions with standard AI assumptions

This appendix provides (i) a diagnostic framework for assessing the degree of wickedness a problem exhibits, and (ii) a detailed enumeration of the structural properties of wicked systems that conflict with standard AI assumptions.

## A.1. From Binary to Spectrum: Diagnosing Wickedness

Treating wickedness as a binary category limits its practical utility: researchers need to assess *how* wicked their problem is, and in what respects, in order to calibrate the depth of methodological response. A diagnostic for wickedness can be built from Rittel & Webber (1973), who identified ten properties that, taken together, distinguish wicked from tame problems. A problem exhibits wickedness to the extent that:

1. **No definitive formulation.** The standard ML pipeline assumes the task can be specified up front—inputs, outputs, loss, data—and then optimised. Wicked problems resist this separation: what the problem *is* only becomes clear through attempts to act on it, so problem-definition and solution-design unfold together rather than in sequence.

2. **No stopping rule.** There is no intrinsic criterion signalling when the problem is solved. Work stops only because of external constraints—time, budget, or patience—not because a solution has been verified complete.

3. **Solutions are good-or-bad, not true-or-false.** There is no loss function whose minimum identifies the correct answer. Solutions are judged against competing values and interests, and different stakeholders will rank them differently with no procedure for adjudicating between them.

4. **No immediate or ultimate test.** Consequences unfold over extended, potentially unbounded time horizons, so no evaluation—held-out set or otherwise—can fully assess a solution. Repercussions may outweigh intended benefits long after implementation.

5. **Every attempt counts.** Interventions are irreversible one-shot operations: there is no sandbox in which to run controlled experiments, and each attempt leaves lasting traces in the world. The iterative trial-and-error on which empirical ML relies is unavailable.

6. **No enumerable solution space.** There is no well-defined hypothesis class or search space to optimise over. Candidate solutions are open-ended, and no procedure can confirm that all possibilities have been considered.

7. **Essential uniqueness.** Despite surface similarities to prior cases, each wicked problem may harbour distinguishing features that override apparent commonalities. Transfer learning intuitions break down: a method that worked previously offers no guarantee for the next instance.

8. **Symptom of another problem.** Any wicked problem can be traced to a deeper one; resolving it at one level may entrench or worsen the higher-order problem of which it is a manifestation.

9. **Explanation determines resolution.** There is no uniquely correct causal account of the gap between the present and the desired state. The choice of explanation—itself a value-laden judgement—determines what counts as a solution, much as the choice of causal graph determines which interventions appear effective.

10. **No right to be wrong.** In science, falsified hypotheses are accepted as part of progress. Planners enjoy no such immunity: they are held accountable for the real-world consequences of their interventions, including those that follow from reasonable but mistaken assumptions.

Table 2 maps the most ML-relevant of these properties to the specific AI assumptions they violate and the research directions they motivate. Importantly, the more of these properties hold, the deeper the methodological response required. Levin et al. (2012) extend the diagnostic to *super-wicked* problems, adding four further markers: time is running out, the actors causing the problem also seek to solve it, central authority is weak, and irrational discounting defers action — a description that closely fits many Anthropocene challenges.

Another particularly useful diagnostic is the two-dimensional framework of (Hisschemöller & Hoppe, 2018), which classifies problems along two axes:

- **Knowledge certainty**: Is the causal structure of the system understood? Are the relevant variables, feedbacks, and future dynamics identifiable and quantifiable?

- **Value consensus**: Do stakeholders agree on objectives, on what counts as success, and on acceptable trade-offs?

Importantly, wickedness also scales with the degree of *coupling to other socio-ecological systems*: each coupling introduces additional knowledge uncertainties (how does the intervention propagate across system boundaries?) and additional value contestations (whose interests in adjacent systems are affected?). Structured problems tend to be self-contained, with tight system boundaries and weak couplings. Fully wicked problems are deeply entangled with other domains—food security, livelihoods, biodiversity, climate, trade—such that intervening in one system inevitably reshapes dynamics in others.

These two dimensions yield four problem types (Figure 2):

1. **Structured problems** (high certainty, high consensus): Well-understood dynamics and agreed objectives. Standard AI methods apply with low risk. *Example*: AI-based coordination of wind turbine orientation to maximise energy output (Howland et al., 2022). The objective is unambiguous, the physics are well modelled, interventions are reversible, and success is directly measurable. The system boundary is tight and couplings to other domains are weak.

2. **Moderately structured (value-contested)**: Causal dynamics are reasonably understood, but stakeholders disagree on goals or acceptable trade-offs. Multi-objective formulations and Pareto-based evaluation become necessary. *Example*: camera-based wildlife monitoring and AI-assisted patrol optimisation for anti-poaching (Duffy et al., 2019; Sandbrook et al., 2021). The technology functions as intended and the causal pathways are well documented. However, camera traps also record people—particularly Indigenous peoples and local communities—and monitoring introduced for ecological purposes can become instruments of surveillance, linked to the militarisation of conservation and the criminalisation of subsistence practices. What registers as improved conservation performance on task-level metrics may simultaneously erode the social legitimacy on which durable conservation depends. The wickedness lies in irreducible value contestation—who defines the problem and whose interests are embedded in system design—rather than in causal opacity.

3. **Moderately structured (knowledge-deficient)**: Objectives are broadly agreed upon, but the system's causal structure or future behaviour is poorly understood. Robustness under deep uncertainty and stress-testing become essential. *Example*: ML-based prediction of ecosystem responses to climate interventions such as reforestation or coral reef restoration. The goal of ecosystem recovery is broadly shared, but the relevant dynamics—species interactions, tipping points, lag effects, responses to novel climate regimes—are characterised by deep uncertainty. Models trained on historical data may fail under conditions with no historical precedent. The wickedness lies in knowledge deficiency: the system's future behaviour cannot be reliably inferred from past observations.

4. **Unstructured (fully wicked)**: Both dynamics and objectives are uncertain, contested, or evolving, compounded by deep entanglement with adjacent systems. The full suite of PCAI principles applies. *Example*: AI-driven precision agriculture for sustainable food production. At first glance, this appears unambiguously beneficial: AI optimises fertiliser, water, and pest management, and performs well on metrics such as yield and input efficiency. However, agriculture is coupled to food security, rural livelihoods, land tenure, water systems, biodiversity, trade, and climate simultaneously—and each coupling introduces both knowledge uncertainties and value contestations. On the knowledge dimension, how AI-optimised farming interacts over decades with soil health, landscape-level biodiversity, and market dynamics driving farm consolidation is poorly understood. Recent evidence suggests that precision agriculture overwhelmingly benefits large commercial operations, with limited demonstrated environmental benefits for the small farms that produce roughly a third of the world's food (Altieri et al., 2024); the systemic effect may be to accelerate consolidation toward industrial monoculture. On the value dimension, agribusiness sees scalable efficiency; smallholders see displacement; ecologists see biodiversity erosion; food sovereignty movements see corporate control of the food system. What makes this *fully* wicked is the interconnectedness: intervening in agricultural efficiency reshapes dynamics across all coupled domains simultaneously. An AI system evaluated at the field level cannot detect the systemic trajectory it helps produce.

This diagnostic operationalises the scaling principle articulated in Section 3: the depth of PCAI engagement should scale with the degree of wickedness. Researchers can use these two dimensions to assess where their problem sits and, correspondingly, which subset of the structural challenges below—and which PCAI responses—are most relevant. (Alford & Head, 2017) offer a complementary typology that adds stakeholder divergence as a third consideration, useful when institutional or political complexity is a primary source of difficulty.

### A.2. Structural Properties of Wicked Systems in Tension with AI

The properties below detail the specific ways in which the sources of wickedness identified above manifest as tensions with standard AI assumptions. Each item identifies a specific way in which common AI methods, evaluation practices, or deployment strategies can fail when applied to wicked systems. The focus is on violations of core technical assumptions rather than downstream ethical or governance outcomes. Together, these properties help explain why AI systems that perform well in controlled or well-specified settings can produce brittle, misleading, or harmful behaviour when embedded in complex and wicked systems.

#### A.2.1. OBJECTIVE & OPTIMIZATION CHALLENGES

These properties arise primarily from low *value consensus*: contested, evolving, or incompatible objectives.

1. **No stable objective function**: Objectives are contested and evolve over time, preventing fixed problem formulation or convergence to a single solution.

2. **Incompatible objectives with no global optimum**: Improvements along one dimension (e.g., efficiency) often degrade others (e.g., equity or resilience), yielding irreducible trade-offs.

#### A.2.2. ENVIRONMENT & DYNAMICS CHALLENGES

These properties arise primarily from low *knowledge certainty*: poorly understood causal structure, feedbacks, and dynamics.

3. **Non-stationary environments**: The data-generating process changes over time, often endogenously in response to the model's own deployment, invalidating assumptions of stable or slowly shifting distributions.

4. **Path dependence and lock-in**: Early interventions constrain future options and are difficult or impossible to reverse, violating assumptions of reversible or correctable decisions.

5. **Nonlinear effects and tipping points**: Small changes can trigger large, abrupt, or irreversible system shifts, undermining local performance guarantees.

6. **Emergent behaviour**: System-level outcomes arise from interactions and feedbacks and cannot be inferred from component-level performance or isolated task metrics.

7. **Deep uncertainty**: Key elements of the system—such as causal structure, relevant variables, future regimes, or outcome priorities—are unknown, contested, or not reliably quantifiable.

#### A.2.3. ACTION & LEARNING CONSTRAINTS

These properties reflect the interaction of both dimensions: acting under combined knowledge and value uncertainty.

8. **No safe exploration regime**: Trial-and-error learning risks real-world, lasting, or irreversible harm, undermining standard exploration assumptions.

9. **Open-ended system evolution**: The relevant state space, action space, objectives, and failure modes cannot be exhaustively specified in advance.

#### A.2.4. DATA & GENERALISATION CHALLENGES

These properties arise primarily from low *knowledge certainty*, compounded by reflexive system dynamics.

10. **Historical data poorly represents the future**: Past data fails to capture emerging regimes, constraints, or feedbacks.

11. **Feedback-contaminated data**: Post-deployment data is shaped by the model's own influence on the system, biasing learning and evaluation.

These properties arise from the combined effect of both dimensions: evaluating interventions whose goals are contested in systems whose responses are uncertain.

12. **No definitive success metric**: There is no agreed-upon way to determine whether an intervention succeeded in contested, long-horizon contexts.

Importantly, many of these challenges have partial analogues in existing AI subfields. For example, non-stationarity is studied in continual and online learning; distribution shift and tail risk are addressed in distributionally robust optimisation; conflicting objectives and value trade-offs appear in multi-objective optimisation and reinforcement learning (RL); delayed consequences and irreversibility are explored in long-horizon and risk-sensitive RL; and endogenous feedbacks are examined in causal modelling and strategic or multi-agent learning. The claim here is not that such tools do not exist, nor that they are irrelevant. On the contrary, these techniques already constitute some of the most promising foundations for AI in complex, high-stakes settings. From a systems perspective, they can be understood as addressing key facets of wicked dynamics albeit often in partial or domain-specific ways. PCAI highlights the need to further integrate these approaches with explicit system-level reasoning. Doing so reframes existing methods as complementary components of a system-aware design and evaluation paradigm. This position paper underscores the importance of sustained research investment in these areas.

## B. Comparison between Human-Centered and Planet-Centered AI Frameworks

PCAI does not replace Human-Centered AI but extends it. HCAI has established essential commitments — to fairness, accountability, safety, and human oversight — that remain necessary under planetary conditions. What PCAI adds is a shift in the unit of analysis, from AI systems and their interactions with individual users or communities to coupled social-ecological systems and their long-run trajectories. Table 3 maps this relationship across thirteen dimensions, showing where the frameworks share common ground and where PCAI introduces genuinely new requirements. The most consequential divergences concern temporal horizon, risk framing, and the role of AI itself: where HCAI asks how AI can serve people responsibly and fairly, PCAI asks whether that service, enacted at planetary scale, contributes to stable and resilient Earth-system futures.

## C. Mini Use Case: System Mapping for AI-Assisted Conservation Enforcement

**Domain:** Biodiversity conservation in protected areas.
**Task-level AI intervention**: AI-enabled surveillance and patrol optimization for anti-poaching.

Recent conservation efforts increasingly deploy AI systems—combining remote sensing and computer vision—to detect poaching activity and optimize ranger patrol routes. These systems are typically evaluated on metrics such as detection accuracy, patrol efficiency, or reductions in poaching incidents. However, as documented extensively in conservation social science, anti-poaching operates within coupled socio-ecological systems characterized by feedbacks between wildlife populations, local communities, rangers, armed groups, and political-economic incentives. In many contexts, AI-enabled enforcement becomes embedded within militarised conservation strategies, with documented long-term consequences for ecological integrity, social legitimacy, and system stability (Duffy et al., 2019; Sandbrook et al., 2021).

### C.1. System Mapping: AI as a Coupled System Intervention

The system boundary includes:

- **Ecological components:** target species populations, habitat quality, and trophic interactions;

- **Human actors:** rangers, local communities, poachers, conservation NGOs, donors, and state agencies;

- **Institutional dynamics:** governance structures, funding mechanisms, and performance metrics;

- **Technological infrastructure:** surveillance hardware, models, data pipelines, and operational protocols.

Key feedback pathways identified through system mapping include:

1. **Enforcement–adaptation feedback:** improved detection alters poacher behavior, potentially increasing displacement, sophistication, or violence;

2. **Militarisation–legitimacy feedback:** Increased surveillance and force reduce community trust, undermining long-term conservation cooperation.;

3. **Resource allocation feedback:** visible enforcement success attracts funding that may crowd out community-based strategies;

4. **Institutional lock-in:** Militarised enforcement increases attacks on rangers, reinforcing justification for further militarisation.

These feedbacks are extensively documented in the literature on militarised conservation (Duffy et al., 2019). This mapping highlights that the primary impact of the AI system lies in shaping how the conservation system evolves together with its coupled impact on human actors, rather than solely in detecting individual poaching events.

### C.2. Theory of Change: AI as a Trajectory-Shaping Intervention

Prior to explicit system mapping, AI-assisted anti-poaching interventions are often guided by a simplified and largely implicit theory of change: improved detection leads to fewer poaching events, which in turn leads to species recovery. While intuitively appealing, this linear causal pathway abstracts away the social, institutional, and political dynamics through which conservation interventions operate, and treats enforcement effectiveness as a sufficient proxy for long-term ecological success. System mapping reveals that this intended causal pathway is neither guaranteed nor exhaustive. Instead of a single dominant mechanism, AI-assisted enforcement activates multiple interacting processes that may reinforce or counteract one another. Under PCAI, AI-assisted enforcement is instead modeled as activating multiple interacting mechanisms, whose relative influence determines long-run system trajectories.

**Core intervention effect.** The AI system reallocates attention, authority, and resources by intensifying surveillance, reshaping ranger practices, and producing data that informs governance decisions and donor priorities.

**Mechanism set A: Short-term deterrence (context-dependent).** In the short run, increased detection may reduce observable poaching activity or displace it spatially. These effects are contingent on limited adaptive capacity and do not address structural drivers of illegal hunting.

**Mechanism set B: Escalation and coercive reinforcement.** As actors adapt, AI-enabled enforcement can justify heightened force, expand surveillance of local populations, and shift ranger roles toward paramilitary functions, reinforcing self-amplifying militarisation dynamics.

**Mechanism set C: Political–economic lock-in.** By privileging quantifiable enforcement outcomes, AI systems shape institutional success criteria and funding flows, entrenching enforcement-centric strategies even when they undermine ecological resilience or social legitimacy.

**Mechanism set D: Social legitimacy erosion.** Expanded surveillance and coercive practices may deepen historical grievances, reduce community cooperation, and weaken informal conservation governance, increasing long-term system fragility.

### C.3. Evaluation Dimensions for Trajectory-Oriented Assessment

Following this system mapping and theory of change, PCAI evaluation focuses on how AI interventions influence long-term system trajectories, rather than point-in-time task performance. Indicative evaluation dimensions include:

- **Ecological resilience:** species recovery under environmental variability and stress;

- **Social legitimacy:** community cooperation, conflict incidence, and trust in conservation institutions;

- **Violence dynamics:** escalation or de-escalation of armed encounters affecting rangers and civilians;

- **Institutional adaptability:** diversity of conservation strategies retained over time;

- **Lock-in risk:** dependence on enforcement-centric approaches and difficulty of reversal;

- **Equity impacts:** distribution of benefits and harms across affected populations.

While many of these dimensions may not be readily quantifiable or reducible to single metrics, making them explicit is nevertheless essential. Explicit articulation of these objectives can shape the choice of modeling paradigm, the form of human–AI interaction, and even the class of interventions considered appropriate. By foregrounding trajectory-level concerns, PCAI encourages discussion of modeling approaches that are developed in close collaboration with domain experts, and supports more meaningful comparison across alternative interventions, including non-AI and hybrid approaches.

Importantly, the need to make such objectives explicit does not imply the existence of a single correct modeling solution. On the contrary, the domains in which PCAI is most relevant are themselves characterized by wicked dynamics, in which no single modeling paradigm can be expected to capture all relevant dynamics. Explicit articulation of trajectory-level concerns therefore serves to structure interdisciplinary modeling processes, constraining assumptions, surfacing uncertainties, and identifying trade-offs that require joint deliberation among ML researchers, domain scientists, and affected stakeholders.

*Table 2.* Wicked System Challenges, AI Limitations, and PCAI-Motivated Research Directions

| Wicked Challenge | How Standard AI Struggles | PCAI-Motivated Research Directions |
|---|---|---|
| **Objective & Optimization Challenges** | | |
| No stable objective function | Assumes objectives can be specified in advance and optimized consistently. This is the case even with AI methods that embed dynamic rewards or loss functions. | Domain generalisation and adaptation, preference learning and adaptive and pluralistic objective representations; methods that support deliberation over objectives and further research on dynamic reward or loss specification. |
| Incompatible objectives with no global optimum | Encodes trade-offs through fixed scalarization or static Pareto formulations, masking irreducible value conflicts. | Trade-off–aware learning and exploration, evolving objective sets, exploratory modeling with multiple objectives, and decision-support tools that surface value conflicts. |
| **Environment & Dynamics Challenges** | | |
| Non-stationary environments | Assumes stable or slowly shifting data distributions; struggles with endogenous change driven by deployment, behavioral adaptation, or policy response. | Continual and regime-aware learning, shift detection, stress-testing under structural breaks, and robustness across plausible future distributions. |
| Path dependence and lock-in | Evaluates actions within bounded horizons, making delayed, irreversible consequences or loss of future options difficult to observe or attribute. | Methods that explore system dynamics and reason about irreversibility, option value, and long-term trajectory selection through simulation and epistemic infrastructures. |
| Nonlinear effects and tipping points | Performance degrades near thresholds where small errors can trigger large or irreversible system responses. | Worst-case analysis, early-warning indicators, and robustness to threshold and tail-risk behavior. |
| Emergent behavior | Task-level validation does not predict system-level outcomes produced by interactions among models, users, and institutions. | System-level evaluation, multi-agent game-theoretic learning, and collective or population-level behavior modeling. |
| Deep uncertainty | Assumes known system structure and probabilistic uncertainty; cannot represent unknown, contested, or unmodellable dynamics. | Decision-making under deep uncertainty, robust satisficing, and epistemic infrastructures for uncertainty aggregation and communication (e.g., expert elicitation, superforecasting). |
| **Action & Learning Constraints** | | |
| No safe exploration regime | Exploration assumes low-cost or reversible errors, inappropriate for high-stakes real-world interventions. | High-fidelity differentiable simulators (digital twins, world models) with foresight, constrained learning, generative AI for scenario generation, and pre-deployment risk analysis. |
| Open-ended and evolving action space | Assumes a fixed and enumerable action space, while real interventions reshape available options and constraints. | Open-ended learning, adaptive action spaces, and methods for reasoning over expanding or evolving intervention sets. |
| **Data & Generalization Challenges** | | |
| Historical data poorly represents the future | Trains on past regimes that omit emerging constraints, feedbacks, or structural change. | Out-of-distribution generalisation, robustness beyond historical fit and hybrid models (e.g. physics-informed neural networks). |
| Feedback-contaminated data | Post-deployment data is endogenous to model, biasing learning and evaluation. | Causal modeling and representation learning, and feedback-aware evaluation. |
| **Evaluation & Benchmarking Challenges** | | |
| No definitive success metric | Presumes agreed-upon success criteria, absent in contested, long-horizon settings. | Multi-criteria, trajectory-oriented, and deliberative/speculative evaluation frameworks. |

*Table 3.* Comparison of Human-Centered AI (HCAI) and Planet-Centered AI (PCAI)

| Dimension | Human-Centered AI (HCAI) | Planet-Centered AI (PCAI) |
|---|---|---|
| **Core guiding question** | How can AI systems be designed and deployed to serve people responsibly, safely, and fairly? | How can AI systems support the long-term stability and viability of coupled Earth systems within which human societies operate? |
| **Ethical baseline** | Anthropocentric: moral priority assigned to human rights, dignity, autonomy, welfare, and social justice. | Relational and ecological: moral concern extends to ecosystems, non-human life, future generations, and life-support systems as conditions for human flourishing. |
| **Primary stakeholders** | Human users, affected communities, workers, institutions, and rights-holders. | Humans, non-human species, ecosystems, future generations, and planetary commons (e.g., climate, biodiversity, biogeochemical cycles). |
| **Scope of responsibility** | Identify, prevent, and mitigate harms to humans arising from AI use, including sociotechnical and institutional impacts. | Account for systemic social–ecological effects, intergenerational impacts, feedbacks, and risks of irreversible or path-dependent change. |
| **Core design objective** | Enable AI systems that are usable, fair, safe, accountable, and aligned with human values and oversight. | Support planetary stability, resilience, and long-term livability by shaping system trajectories toward desirable futures. |
| **Unit of analysis** | AI systems and their interaction with users, organizations, and sociotechnical contexts. | Coupled social–ecological systems, intervention pathways, and multi-scale dynamics over time. |
| **Definition of progress** | Improvements in task performance, usability, fairness, accountability, transparency, and human capabilities. | Reduced systemic risk, preserved ecological integrity, resilience under change, and maintenance of future options. |
| **Temporal horizon** | Primarily near- to medium-term impacts surrounding design, deployment, and use. | Explicitly long-term and intergenerational, accounting for cumulative, delayed, and irreversible impacts. |
| **Risk framing** | Algorithmic and sociotechnical risks affecting humans, such as bias, misuse, privacy loss, exclusion, and unsafe automation. | Systemic and planetary risks, including feedback loops, rebound effects, correlated failures, and tipping points. |
| **Evaluation logic** | Combination of task-level benchmarks, user studies, and post hoc impact assessments. | Trajectory-oriented evaluation, including stress-testing under non-stationarity, scenario analysis, and system-level probes. |
| **Role of AI** | Human-centered decision support, augmentation, and automation under meaningful human control. | Epistemic and stewardship-oriented: sense-making, foresight, scenario exploration, and support for collective judgment under deep uncertainty. |
| **Knowledge foundations** | Computer science, human-computer interaction, ethics, law, psychology, and social sciences. | Systems science, Earth-system science, ecology, sustainability science, political economy, and plural epistemologies. |
| **Governance model** | Institutional compliance, accountability mechanisms, and oversight focused on specific deployments. | Planetary stewardship: governance oriented toward long-horizon coordination, boundary-setting, reversibility, and adaptive control across scales. |

