# OpenReview forum: "Position: AI Must Become Planet-Centered, Not Just Human-Centered"
_ICML.cc/2026/Position_Paper_Track — ICML 2026 Position Paper Track regular_

### Official Review · Reviewer_tHLC · 2026-02-21

**Significance:** 3
**Argument Clarity:** 2
**Rating:** 5
**Confidence:** 4

**Questions:**

- How should researchers systematically determine whether a given problem setting qualifies as a wicked system, beyond the illustrative examples provided, and can the authors propose a more general or operational framework for this classification?


 - Could the authors introduce a clearer taxonomy of Anthropocene‑era ML systems that contribute to environmental degradation, including guidance on how proxy objectives should be formalized, optimized, and stress‑tested within such systems?


 - The title appears to suggest a strong contrast between planet‑centered and human‑centered AI, while the paper later argues for complementarity between these paradigms; would the authors consider clarifying this relationship to avoid potential overstatement or misinterpretation of their position?


 - How do the authors envision practitioners navigating the trade‑offs implied by robustness‑oriented or minimax objectives, particularly in cases where such formulations may lead to degraded task‑level performance or reduced usability, and when are these trade‑offs justified?

**Alternative Views Section:**

Yes

**Compliance With Llm Reviewing Policy A Conservative:**

Affirmed.

**Discussion Potential:**

3

**Final Justification:**

The rebuttal has clarified my concerns and I have increased my score leaning towards accept.

**Paper Summary:**

This position paper argues that prevailing AI paradigms, such as human‑centered, responsible, or sustainable AI, are inadequate for addressing planetary‑scale challenges characterized by systemic risk, non‑stationarity, and deep uncertainty. It proposes Planet‑Centered AI (PCAI) as a design philosophy and research agenda that treats Earth’s coupled socio‑ecological systems, their feedbacks, and long‑term trajectories as first‑order design concerns, rather than as externalities inferred through human outcomes. The paper diagnoses recurring failure mechanisms of current AI approaches in “wicked” systems, including technical misalignment, narrow value framings, impact blind spots, and risk amplification, and shows how these can exacerbate systemic instability even when models perform well on standard metrics. PCAI is articulated across the AI lifecycle. emphasizing systems mapping, theory‑of‑change reasoning, robustness under deep uncertainty, trajectory‑oriented evaluation, and ongoing monitorability. The authors advance a falsifiable claim that AI systems optimized without explicit consideration of systemic feedbacks are more likely to worsen, rather than mitigate, planetary instability, and conclude with a concrete call to action for researchers, institutions, funders, and the ML community to realign incentives, methods, and evaluation norms toward long‑term planetary stewardship.

**Position:**

Yes

**Position In Title:**

Yes

**Related Work:**

2

**Strengths And Weaknesses:**

Strengths:

 - The paper addresses an important and timely area by articulating the need to rethink prevailing AI paradigms in light of planetary‑scale challenges.
 - Its central arguments are presented in a balanced and well‑reasoned manner, with claims consistently grounded in prior literature and clearly aligned with the stated position.
 - The discussion of alternative viewpoints is handled thoughtfully: opposing perspectives are treated as credible rather than dismissed, and the authors explicitly position their proposal as complementary to, rather than in conflict with, existing frameworks.
 - The paper offers an explicit and structured call to action, outlining concrete implications and responsibilities for different segments of the machine learning community, including researchers, institutions, funders, and venues.

Weaknesses:

- The paper does not provide a clear or systematic framework for identifying or characterizing “wicked” systems. While several illustrative examples are discussed, these remain selective and non‑exhaustive. In the absence of a more principled characterization, it is unclear how researchers or practitioners should reliably determine whether a given problem setting qualifies as a wicked system, which may limit the practical applicability of the proposed paradigm.


- Relatedly, the paper would benefit from a more structured taxonomy of Anthropocene‑era ML systems that contribute to environmental degradation, along with a clearer formalization of the proxy objectives or system‑level indicators that could be used for optimization or stress‑testing. Without such abstractions, it remains difficult to translate the conceptual arguments into concrete methodological guidance for model design and evaluation.


- The title asserts that AI should not be human‑centered; however, the body of the paper repeatedly emphasizes that Human‑Centered AI and Planet‑Centered AI are complementary rather than mutually exclusive. This creates a tension between the framing in the title and the more nuanced position articulated in the text, and the title may therefore overstate the degree of opposition to existing human‑centered paradigms.


- While the paper calls for explicit consideration of trade‑offs, it does not sufficiently delineate their practical consequences. For example, robustness‑oriented or minimax objectives, while appealing from a systemic risk perspective, may substantially degrade task‑level performance or usability in practice. A clearer discussion of these trade‑offs, including when such sacrifices may or may not be justified, would strengthen the argument and help situate PCAI relative to existing optimization paradigms.

**Support:**

2

---

> ### Author Rebuttal · Authors · 2026-03-30
>
> We thank the reviewer for this detailed and analytically demanding engagement with the paper. We commit to making revisions addressing the points raised.
>
> **Systematic framework for wicked problems.** We agree this is essential. Appendix A will provide two tools: We include Rittel & Webber's (1973) ten properties of wicked problems as a standalone diagnostic resource, alongside a two-dimensional framework adapted from Hisschemöller & Hoppe (1996) that classifies wicked problems by knowledge certainty and value consensus, yielding four quadrants relevant to AI, each with a worked example:
>
> - Structured: Well-understood dynamics and agreed objectives. Examples include an agent that plays chess, or one that coordinates wind turbine orientation to maximise energy output (Howland et al., 2022, Nature Energy). Objectives are unambiguous, dynamics are well-modelled, standard AI applies.
> - Value-contested: Dynamics understood, goals disputed. E.g., camera-based conservation monitoring (Duffy et al., 2019, Biological Conservation; Sandbrook et al., 2021, Conservation Science and Practice): technology detects poaching as intended, but camera traps also record Indigenous peoples and local communities. Monitoring introduced for ecology has become an instrument of surveillance, characterised as the militarisation of conservation. What is contested is not whether the technology works, but e.g. who it serves, who bears its risks, and whether detection metrics justify eroding the social legitimacy on which durable conservation depends.
> - Knowledge-deficient: Goals shared, dynamics deeply uncertain. E.g., ML prediction of ecosystem responses to reforestation, where species interactions, tipping points, and novel climate regimes cannot be reliably inferred from historical data.
> - Fully wicked: Both dynamics and objectives uncertain. E.g., AI-driven precision agriculture. On knowledge uncertainty: how AI-optimised farming creates feedback loops between efficiency, monoculture, and ecological resilience is unknown. On value contestation: agribusiness sees yield optimisation; smallholders see displacement and loss of autonomy; ecologists see monoculture lock-in and biodiversity erosion.
>
> Importantly, wickedness scales with coupling, each coupling introducing additional uncertainties and value contestations. This diagnostic operationalises the scaling principle in Section 3: the depth of PCAI engagement should scale with the degree of wickedness. The appendix will clarify how PCAI mechanisms map to each quadrant: e.g. system mapping to surface contestedness, foresight and hybrid intelligence to navigate deep uncertainty, and the full suite for fully wicked.
>
> **Taxonomy and proxy objectives.** The failure mechanisms in Section 2 provide a natural taxonomy of how AI contributes to systemic degradation: technically misaligned solutions, framing that is too narrow for the system at hand, impact assessment that does not consider system entanglement, risk amplification through scale, absence of principled direction reinforcing our falsifiable claim in 3.5. On the reviewer's important question about formalising objectives: PCAI argues that objectives should emerge from system mapping and theory of change, the first stage of PCAI. This process identifies which dimensions of the system are consequential, where may be the leverage points and which proxies risk diverging from the values they represent. Table 2 then provides guidance, mapping wicked properties to the AI assumption it violates and the methodological response it demands. For instance, non-stationarity calls for out-of-distribution stress-testing; feedback contamination calls for protocols that account for the model's influence on the data. A definitive taxonomy remains difficult because failure modes are entangled, but we see this as a foundation to be extended.
>
> We note that PCAI does not focus exclusively on environmental objectives, it centres systems thinking, of which humans and environment are inseparable. We foreground environmental examples given that only 16–26% of AI ethics guidelines address these (Section 2.2.3).
>
> **Title.** We chose "not human-centered" for provocation per ICML guidelines, but have been pondering "not just human-centered," which better reflects the complementarity. We are happy to adopt this if reviewers consider it more appropriate.
>
> **Trade-off costs.** We agree robustness may degrade task performance and do not claim these trade-offs are costless. PCAI aims to make them visible: the Pareto framework surfaces trade-offs for deliberation rather than hiding them in a single objective, and system mapping makes explicit the interconnections through which narrow optimisation produces unintended consequences. See response to Reviewer Exqr (Weakness 2): the difficulty of evaluation is inherent to wicked problems. We have proposed mechanisms to address this but are candid that this is a vision under construction, one we hope the community will extend.

---

> > ### Author Rebuttal · Reviewer_tHLC · 2026-04-02
> >
> > The rebuttal has clarified my concerns and I have increased my score leaning towards accept.

---

### Official Review · Reviewer_K5ig · 2026-03-12

**Significance:** 3
**Argument Clarity:** 3
**Rating:** 5
**Confidence:** 3

**Questions:**

The paper was very well written so I do not have any major clarifying questions. However, I would greatly appreciate it if the authors could please address my comments made in the "weaknesses" section above, concretely:
- Concrete examples of AI risks not well captured by exisiting frameworks
- More context (ideally supported by citations) for the alternative views presented

**Alternative Views Section:**

Yes

**Compliance With Llm Reviewing Policy A Conservative:**

Affirmed.

**Discussion Potential:**

3

**Paper Summary:**

This position paper argues that the current paradigms for mitigating ethical, social, and technical risks of AI development do not adequately take into account planetary-scale socio-ecological systems and their long-term trajectories. Existing paradigms tend to only consider these effects indirectly, e.g. human-centered AI considers environmental instability indirectly as it will impact the well being of humans in the long run, while technology accelerationist views tend to argue that technological innovation itself will be a solution to any unintended downstream effects of AI progress. The position paper clearly lays out these current positions and how they differ in emphasis and scope compared to the planet-centered AI (PCAI) view that the paper argues for. Specifically, a PCAI-ist approach to AI development is grounded in systems and planet-level thinking (of which humans certainly are one of several factors), and views the problem of AI as a "wicked" system that inherently exhibits ambiguity, non-stationarity, and deep uncertainty. The paper presents the PCAI position in detail, and concludes with a call to action that clearly articulates what various actors including researchers, educators, and organizations can do to act upon the presented framing.

**Position:**

Yes

**Position In Title:**

Yes

**Related Work:**

2

**Strengths And Weaknesses:**

Overall, this position paper was an enjoyable read with a clearly articulated vision for the future of AI ethics. I currently lean towards acceptance and describe my identified strengths and weaknesses in the following.

**Strengths:**
- The paper is generally very well written and easy to follow. The paper is well organized, clearly positions itself against other frameworks for AI risk, and draws connections to several other relevant domains. Figure 2 and Table 1 are especially helpful for understanding the core mechanisms and actions argued for in the position paper.
- A planet-centric view of AI risks operates at a high level of abstraction which has the benefit of encapsulating long-term systems-level risks often not considered explicitly by e.g. the human-centric AI community. The downside of this level of abstract is that it can be difficult to operationalize but I believe that the paper does a good job at conveying how it can be implemented more concretely (again, Table 1 presents this very clearly).

**Weaknesses:**
- I believe that the paper would benefit from emphasizing a couple of concrete examples of systems-level risks not well captured by current frameworks for AI risk. The paper currently treats these risks at a pretty abstract level, so providing concrete examples might make it easier for readers to contextualize.
- The alternative views section presents views that are intuitive to understand as someone familiar with the area, but they are pretty sparse with details and lack citations. It would be helpful to rewrite the section to provide more context to these views and make the paper more self-contained.

**Support:**

3

---

> ### Author Rebuttal · Authors · 2026-03-30
>
> We thank the reviewer for the generous assessment and constructive suggestions, which have improved our paper.
>
> **Weakness 1 (examples of systems-level risks).** We will highlight categories of AI risk not well captured by existing frameworks, rooted in systems thinking:
>
> - Risks from entanglement across system boundaries. Current AI risk frameworks assess systems largely in isolation. But when an AI intervention in one domain, e.g. agricultural efficiency, propagates into food security, water systems, biodiversity, and rural livelihoods simultaneously, the risks emerge from the couplings between systems rather than from any single system alone. These cross-boundary risks are invisible to many AI risk frameworks. Our worked examples (autonomous vehicles, Aadhaar, precision agriculture) each illustrate this pattern.
> - Risks from the depth of intervention in the system. Systems thinking distinguishes between intervening at the level of parameters (making an existing process faster or cheaper) and intervening at the level of system goals or structure (Meadows, 2008). Most AI operates at the parameter level (optimising within existing dynamics) which can reinforce trajectories that are themselves unsustainable. Existing risk frameworks do not assess where in the system's causal structure an AI intervention acts, or whether that intervention reinforces or redirects systemic dynamics.
>
> These examples will help the reader contextualise systemic risk. However, a full mapping of these risks exceeds position-paper scope. Instead, we treat PCAI as a vision in progress, one we hope the community will expand, as HCAI has been.
>
> To further contextualise systems thinking, we will strengthen the engagement with complex systems throughout. In the introduction, we plan to trace how stressors cascade: climate change warms waters, shifting predator populations, enabling jellyfish blooms that clog coastal power plant intakes and trigger energy instability (Purcell et al., 2007, Marine Ecology Progress Series). We will link to evidence that over half of known human infectious diseases can be aggravated by climate change through more than a thousand systemic pathways (Mora et al., 2022, Nature Climate Change).
>
> Understanding these cascades before designing AI interventions is essential, yet AI practice currently lacks systematic methods for doing so. Methods for system mapping exist in adjacent fields (policy analysis, systems engineering) but have not been imported into AI practice. A key contribution of PCAI is to establish these as design preconditions within the AI lifecycle. We will cite recent work documenting unintended consequences of AI in sustainability through systemic pathways (Gohr et al., 2025, Nature Sustainability; Vinuesa et al., 2020, Nature Communications; Adams et al., 2026, Nature Computational Science).
>
> The revision of our work will add further illustrative examples (see response to Reviewer Exqr) and a diagnostic framework for wickedness in Appendix A with four sustainability examples across the spectrum (see response to Reviewer tHLC).
>
> **Weakness 2 (more context for alternative views).** We will strengthen all subsections of Section 4:
> - 4.1 (HCAI as Sufficient): Vinuesa et al. (2020, Nature Communications) find AI may enable 79% of SDG targets but inhibit 35%, yet acknowledge that cross-target interactions remain poorly characterised. PCAI responds to their call for methodologies that assess systemic interactions.
> - 4.2 (AI Safety): PCAI shares AI safety's concern with alignment and unintended consequences (Russell, 2019) but extends the analysis from model behaviour to system-level trajectories — from "does the model do what we asked?" to "does this intervention produce the trajectory we intended?"
> - 4.3 (Systemic Reasoning Outside AI's Scope): Adams et al. (2026, Nature Computational Science) review 25 GenAI deployments in LMICs and find practitioners shifted from tech-solutionism to socio-technical approaches as "problems tended to be nested." Systemic reasoning is not optional but an emergent requirement of deployment.
> - 4.4 (Technological Progress as Primary Solution): Toyama (2010; 2011; 2015) proposes a "Law of Amplification" based on a decade of field research at Microsoft Research India across 50+ projects: technology amplifies existing conditions — accelerating progress where dynamics are favourable, widening gaps where they are not. Randomised trials found PCs supplementing good teachers helped, but PCs substituting for teachers hurt. His analysis of whether technology can end poverty (Toyama, 2010, Boston Review) concludes it cannot, not because it fails technically, but because it magnifies pre-existing conditions, including inequality. The amplification thesis predicts exactly the dynamics PCAI identifies: AI optimised for task performance will amplify existing trajectories unless objectives are deliberately reoriented.
>
> We commit to making these changes for the camera-ready version.

---

> > ### Author Rebuttal · Reviewer_K5ig · 2026-04-03
> >
> > Thank you for the detailed response! This addresses my concerns. I will keep my score (Accept).

---

### Official Review · Reviewer_Exqr · 2026-03-12

**Significance:** 4
**Argument Clarity:** 3
**Rating:** 5
**Confidence:** 3

**Questions:**

(1) The paper motivates the need to change how AI is approached and evaluted, as a follow up, what are some scenarios in which changes in research or AI responses exacerbate global “wicked” problems? Are there current practices that have clearly demonstrated adverse global effects but positive human centric performance?

(2) As discussed in weakness (2) how would these metrics help determine which models "win" at predicting climate change? Is there a way to understand if the metrics are too narrow for the wicked problems?

**Alternative Views Section:**

Yes

**Compliance With Llm Reviewing Policy A Conservative:**

Affirmed.

**Discussion Potential:**

4

**Final Justification:**

My recommendation was accept and it remains so. My concerns have been adequately addressed.

**Paper Summary:**

This paper introduces and argues for PlanetCentered AI (PCAI), which they define as a design philosophy and research agenda that reorients the goals for AI, rather than optimizing for specific tasks using a specified set of metrics, the goal should be to optimize for shaping the future.

**Position:**

Yes

**Position In Title:**

Yes

**Related Work:**

4

**Strengths And Weaknesses:**

Strengths:
(1) Well written, cohesive, comprehensively discussing alternative perspectives throughout the paper, discusses issues with the existing goals and benchmarks of AI.

(2) The paper does a good job of drawing information and insights from a wide set of disciplines such as economics and anthropology.

Weaknesses:
(1) The paper discusses examples of scenarios very briefly. It is understood that "biodiversity loss and the destabilization of life-support systems unfold gradually”, but with the argument for methods that benchmark “anticipating indirect, cumulative, long-horizon, and intergenerational effects” it would strengthen the paper to discuss such examples in full, perhaps demonstrating how short term human centric decisions, which improve efficiency in narrow system boundaries, may contribute to planet-scale longer term issues.

(2) The paper proposes 3 new ways to perform evaluation (section 3.3)  (a) Pareto frontiers to see the tradeoffs between other metrics like efficiency vs environmental effect, (b) Using simulations to probe risk, and (c) comparison against non-ML baselines.
At the same time the paper argues that current AI frameworks fail at planetary-scale wicked problems due to their inability to handle deep uncertainty or predict long term effects from short term decisions.
But it is not clear how we can be certain that these proposed evaluation techniques are still not too narrow for wicked problems.

**Support:**

3

---

> ### Author Rebuttal · Authors · 2026-03-30
>
> We thank the reviewer for the thoughtful evaluation. The questions have sharpened the paper’s engagement with evaluation and trade-offs.
>
> **Weakness 1 (fuller examples).** The revision will add three examples demonstrating how improving narrow metrics can contribute to systemic harm.
>
> - Autonomous vehicles: AI designed for environmental efficiency produces a 21% reduction in operational emissions but an 8% net increase in lifecycle GHG due to rebound effects due to induced demand, modal shift from transit, and empty vehicle miles (Onat et al., 2023, Nature Communications; Taiebat et al., 2019, Applied Energy).
> - Social media recommendation: an audit shows algorithms amplify hostile content users don't prefer, through feedback loops invisible to engagement metrics (Milli et al., 2025, PNAS).
> - Biometric welfare: India's Aadhaar system uses biometric matching to authenticate beneficiaries collecting food rations, designed to serve the undocumented poor but evaluated on eliminating ghost and duplicate identities. A large-scale evaluation found that while corruption fell, approximately half of the reduction in disbursals came from excluding legitimate beneficiaries, with 1.5–2 million people losing access (Muralidharan et al., 2025, Review of Economics and Statistics). Biometric failure rates of 7–17% (Khera, 2017, Economic and Political Weekly) impacted manual labourers and the elderly, leading to documented starvation deaths  (Khera, 2019, Economic Sociology).
>
> **Question 1 (adverse global effects).** In each case, the AI succeeds on the metrics defined (efficiency, engagement, identifying duplicate identities); while the system-level outcome is the opposite of what was intended. A review of 25 GenAI deployments in LMICs corroborates this: pro-social AI applications encountered wicked dynamics (e.g. contested objectives, reflexive system responses, and nested problems (Adams et al., 2026, Nature Computational Science)). Our paper positions these risks as scaling with wickedness and includes a diagnostic framework (see response to Reviewer tHLC). These adverse effects underscore the need to understand the system and its dynamics before designing a solution. Thus, system mapping and theory of change are central proposals of PCAI.
>
> **Weakness 2 (evaluation adequacy).** We acknowledge this is an inherent property of wicked problems: there is no definitive test of a solution (Rittel & Webber, 1973, Policy Sciences, property 4). PCAI addresses it through three mechanisms. First, the wickedness diagnostic (see response to Reviewer tHLC) calibrates evaluation depth to problem complexity. Second, PCAI proposes that the first stage of the AI lifecycle for wicked problems should be a full system mapping — variables, interconnections, feedback loops, leverage points — and a theory of change. This surfaces challenges with current evaluation metrics and deployment risks such as rebound effects or failures to account for interconnectedness. Third, monitorability and revisability embed ongoing assessment into the deployment lifecycle precisely because upfront evaluation cannot be guaranteed sufficient.
>
> **Question 2 (narrow metrics).** The reviewer's own example, predicting climate change, illustrates this well. For wicked problems with deep uncertainty, evaluation cannot reduce to e.g. the highest R². Several structural properties of wicked problems apply: historical data does not capture future conditions; small changes can trigger large-scale nonlinear shifts; and deep uncertainty means key parameters and even system structure may be unknown. These properties tell us which evaluation approaches are needed: out-of-distribution generalisation tests, benchmarks that probe nonlinear and chaotic behaviour, and robustness assessment across sources of uncertainty (e.g., degrees of warming, ice-sheet dynamics). This is where PCAI emphasises moving beyond single-score ranking. The Pareto-front framework makes trade-offs visible rather than collapsing them into a single metric. A system map and theory of change surface further insights. Climate change is entangled with human actions (e.g. energy systems) so the challenge is not just prediction but foresight: understanding how different actions lead to different futures. Evaluation must therefore assess not which model best fits the past, but which model most usefully illuminates the space of possible Earth trajectories, so that humans may understand what is needed for climate change mitigation. Monitorability highlights evaluation evolving as the climate system and human responses to it change. We agree with the reviewer that metrics may always be reductionist in these contexts. We do not claim planetary wellbeing can be captured through easily quantifiable metrics, proxies will be necessary, as will speculative analysis of impacts. PCAI aims to foreground systemic impact as one of the grand challenges of AI evaluation.
>
> We commit to these changes for the camera-ready version.

---

> > ### Author Rebuttal · Reviewer_Exqr · 2026-04-02
> >
> > My recommendation was accept and it remains so. My concerns have been adequately addressed.

---

### Official Review · Reviewer_ayrd · 2026-03-13

**Significance:** 3
**Argument Clarity:** 2
**Rating:** 5
**Confidence:** 4

**Questions:**

Nothing in addition to what I already wrote.

While I am always open to discussion and changing my mind, in this case, the lack of sufficient technological grounding of the position limits the impact of the paper in my opinion. While this can be certainly addressed, I am afraid that it would be difficult to do so in the short time of the discussion period.

**Alternative Views Section:**

Yes

**Compliance With Llm Reviewing Policy A Conservative:**

Affirmed.

**Discussion Potential:**

3

**Final Justification:**

I upgraded my scores, as the authors provided satisfactory replies to my comments in the rebuttal.

**Paper Summary:**

This paper is positioned in favor of planet-centered AI, as opposed to the current human-centered AI. The paper argues that several current societal challenges are not isolated but intertwined by complex interactions, and that human-centered objectives that are shortsighted may result in unintended negative effects when deployed globally.

The paper covers the limitations of current AI paradigms, in regard to the so-called "wicked problems", which are prevalent when looking at the Earth as a system, narrow framing of the problems, blind spots and risk amplification among others. After that, it proposes the new planet-centered paradigm by proposing shared goals, system-aware models and monitoritability for the definition of AI systems. Alternative views include include 1) technological progress as the main goal, 2) human-centered AI as sufficient for planetary wellbeing, and 3) that the topic is outside of the AI reach.

**Position:**

Yes

**Position In Title:**

Yes

**Related Work:**

2

**Strengths And Weaknesses:**

The authors state their position sensibly and provide credible motivation at high level. As the authors claim, the topic is of high relevance and even urgency. It is highly likely that the topic would generate relevant discussion, and it would be beneficial for everybody that the ICML and ML community in general would engage in the topics exposed.

The position paper, however, presents insufficient low-level details and support that may lead to focusing and grounding of the discussion. For example, while the paper identifies the lack of shared goals as a limitation, it does not propose a clear and detailed set of potential shared goals, but rather presents a discussion of possibilities at high level. In the same manner, the paper characterizes the Earth global dynamics as a wicked problem, which I find illustrative and interesting, but fails to identify specific research topics that could address such problems, or which specific areas should be promoted. Summing up, while the high-level discussion of the paper certainly has value, I think that grounding abstract thinking into specific actions, related to technological development, is essential in a position paper to set things in motion and be influential.

**Support:**

3

---

> ### Author Rebuttal · Authors · 2026-03-30
>
> We thank the reviewer for recognising the high relevance and urgency of the topic. The feedback has helped us strengthen the paper's concreteness substantially.
>
> **Actionable technological grounding.** The paper contains two sections aimed at grounding PCAI. Table 1 presents a call to action with directions for applied and foundational researchers, institutions, conferences, education, and governance, including extending impact assessment to embed lifecycle systemic risk analysis, developing impact datasheets, extending AI ethics frameworks to planetary responsibility, emphasising explicit system mapping, and developing ML methods for foresight and evolving complex systems, among others.
>
> Table 2 (Appendix) links structural properties of wicked problems to the standard assumptions they violate in mainstream AI and to the research directions PCAI motivates. For example, open-endedness, historical data poorly representing the future, and feedback-contaminated data. Many of these challenges have partial analogues in existing ML subfields: non-stationarity in continual learning, distribution shift in distributionally robust optimisation, conflicting objectives in multi-objective RL. PCAI argues both that these techniques deserve sustained research investment because they address dynamics central to planetary-scale challenges, and that the complexity of the problems themselves (i.e. their wickedness) must be understood first to determine which technical foundations are most needed and where current methods fall short. We will emphasize these two resources in the revised version of the paper.
>
> For the camera-ready version, we commit to the following concrete changes:
>
> - Name specific AI assumptions that wicked dynamics violate in the main text, with more explicit links to the technical material in the Appendix. Examples include stationarity, sample independence, decomposability, local smoothness, static objectives and closed-world formulation.
> - Open Appendix A with a diagnostic framework inspired by (Hisschemöller & Hoppe, 1996, Policy Sciences) categorising wicked problems along two dimensions: knowledge certainty and value consensus, from structured (tame) to fully wicked. Each quadrant is illustrated with a worked sustainability example (wind farm coordination, conservation monitoring, ecosystem prediction, precision agriculture) specifying which PCAI measures apply. See response to Reviewer tHLC for details.
> - Add a technical case study in Section 3.2 illustrating foresight in a fully wicked domain: policy making for sustainability transitions. This is the domain addressed by IPCC reports, where simulations and world models underpin future scenario analysis. Recent work has applied multi-agent RL to explore the space of possible climate policies, building foresight tools for policymakers (Rudd-Jones et al., 2025, Environmental Data Science; Filatova et al., 2025, PNAS). Foresight handles both knowledge uncertainty and value contestedness (the two dimensions that diagnose full wickedness) making it a critical technical foundation for planetary-scale challenges.
> - Strengthen Section 4 (Alternative Views) with new citations engaging named proponents, see response to Reviewer K5ig for details.
> - Demonstrate additional systems thinking concepts in the text, helping the reader navigate concepts such as systemic risk, leverage points, feedback loops, etc. See response to reviewer K5ig.
>
> **Concrete examples.** For the camera ready version we commit to grounding the AI failure mechanisms that we identify in Section 2 with additional documented cases: autonomous vehicles producing net lifecycle emission increases through rebound (Onat et al., 2023, Nature Communications); recommendation algorithms amplifying content users don't prefer (Milli et al., 2025, PNAS); and biometric welfare authentication where half of measured savings came from excluding legitimate beneficiaries, with documented starvation deaths (Muralidharan et al., 2025, Review of Economics and Statistics). See response to Reviewer Exqr.
>
> **Shared goals.** The reviewer notes the paper does not propose specific shared goals. This is deliberate: PCAI argues that international agendas (the SDGs, planetary boundaries, social foundations) should serve as the guiding principles determining which technical foundations to develop. PCAI asks where AI can provide the greatest leverage for systemic change (Meadows, 2008), and what technical challenges — reasoning under deep uncertainty, multi-stakeholder objective specification, long-horizon evaluation — currently prevent it from doing so. A comprehensive system map, the first stage of the PCAI pipeline, can identify such leverage points while surfacing risks of unintended consequences.
>
> We see PCAI as a framing to be extended, much as HCAI evolved from a position into a research programme. Its core contribution is embedding systems thinking into the AI pipeline, from problem setting to deployment.

---

> > ### Author Rebuttal · Reviewer_ayrd · 2026-04-03
> >
> > I thank the authors for their insightful rebuttal, that gave answer to my questions.
> >
> > I acknowledge that I may have been a bit harsh on my comment about low-level details, as the authors argued pointing to several specific parts of their submission.
> >
> > I appreciate that they committed to add very specific cases, that they also comment with detail in the rebuttal. Anchoring their discussions on this highly relevant topic with concrete examples would significantly strengthen the paper.
> >
> > The authors convincingly argued against proposing shared goals, that was another of my comments, but rather focusing on other aspects in the rebuttal.

---

### Decision · Program_Chairs · 2026-04-30

**Decision:**

Accept (regular)

**Comment:**

Reviewers agreed that this paper presents a clear and well-supported claim that is timely and even urgent. The paper draws evidence from multiple disciplines to present well-structured and convincing evidence. Reviewers felt that some specific examples and structured taxonomy for certain concepts would improve some limitations in the paper’s original draft, and were satisfied by the authors’ commitment to make these changes in the camera-ready version.